# RetroAux: Boosting Retrosynthesis via Molecular Property-Aware Learning

## Abstract

Retrosynthesis remains a critical task in both drug discovery and organic synthesis. Current methodologies in this field predominantly rely on purely data-driven paradigms, where models are expected to autonomously learn reaction patterns from extensive retrosynthesis datasets without incorporating established chemical knowledge. To address this limitation, we introduce **RetroAux**, a framework injected with molecular property knowledge to enhance existing approaches to achieve significant performance enhancements. Specifically, throughout the entire retrosynthesis pipeline from reaction rule acquisition to reactant molecule generation, our methodology systematically integrates molecular property knowledge (e.g., functional groups, structure) as both chemical priors and foundations, thereby enhancing retrosynthesis prediction reliability. Our *knowledge-driven* framework can be seamlessly integrated with multiple existing data-driven methods and improve their performance stably. Experimental results demonstrate that it enhances various existing data-driven retrosynthesis models with average top-1 accuracy improvements of **2.22**% without retraining origin models, signifying a paradigm evolution in retrosynthesis from purely data-driven approaches to knowledge-driven methodologies.

## 1 Introduction

Retrosynthesis(Long et al., 2025; Torren-Peraire et al., 2024), a fundamental task in chemistry, aims to infer feasible synthetic routes for target molecules through intricate chemical transformations. As the cornerstone of retrosynthesis, single-step retrosynthesis that focuses on synthesizing the target molecule through a reaction step has attracted significant interest (Maziarz et al., 2025), with continuous progress being made in recent years (Somnath et al., 2021).

Contemporary retrosynthesis methods (Zhong et al., 2024; Guo & Schwaller, 2025) concentrates on designing sophisticated architectures to learn more reaction rules purely from reaction data, and template-free methods (Jiang et al., 2023) have recently achieved state-of-the-art (SOTA) performance among them. Nevertheless, they only model correlations between atom-level tokens in reactions due to the lack of chemical knowledge guidance. This results in models that primarily learn statistical patterns of atomic arrangements rather than emulating chemical reasoning processes. Furthermore, these models output molecules through atom-by-atom generation without holistic molecular comprehension, which leads to output reactions chemically invalid. Through careful analysis, we identify two fundamental limitations in current approaches: (1) Overreliance on data-driven pattern learning without reaction-relevant molecular property knowledge as chemical priors, (2) unconstrained model outputs lacking chemical knowledge constraints.

Chemical reactions originate from molecular interactions, where molecular properties serve as the fundamental basis determining reaction feasibility, pathways, and efficiency (Hammond, 1955; Kolb et al., 2001; Noyori, 2002). Therefore, it is natural and critical to integrate the knowledge of molecular chemical properties into retrosynthesis models. Chemists typically infer possible reactions and reactants by analyzing correlations between product and reactant chemical properties (Corey, 1967). Inspired by this, our method introduces molecular chemical knowledge as *a prior* to understand chemical reaction mechanisms in retrosynthesis. During inference, we constrain the retrosynthesis model outputs through the molecular properties. This establishes a paradigm shift from merely learning correlations between molecular SMILES strings (Weininger, 1988) or graph

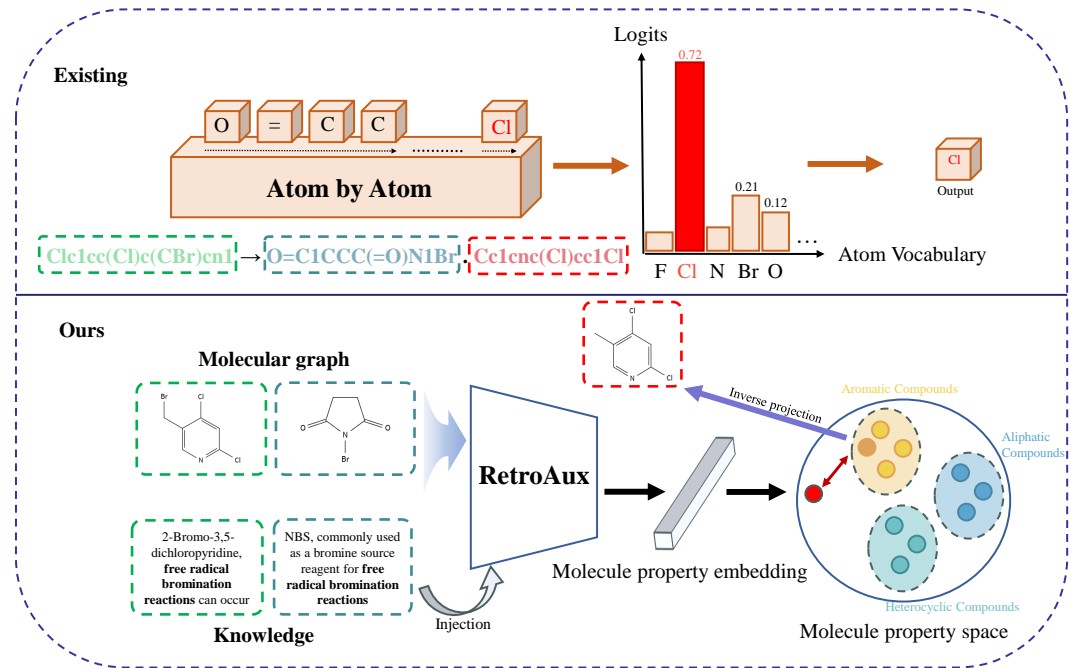

Figure 1: **Standard retrosynthesis autoregressive modeling (Existing) *vs* our proposed RetroAux (Ours).** (a) AR applied to SMILES: sequential atom token generation from left to right, atom by atom, no chemical knowledge; (b) RetroAux directly generates molecular property vectors guided by chemical knowledge, subsequently outputting complete molecules through inverse projection in chemically constrained spaces.

structures in reactions, to explicitly model interdependencies among the chemical properties of participating molecules. Besides, our model operates at the molecular level, reducing dependency complexity. Unlike atom-level methods that must model intricate dependencies among dozens of atoms per reaction, our approach only needs to model relationships between participating molecules, significantly simplifying the task, thereby alleviating the combinatorial complexity associated with sequential atom reconstruction. By generating entire molecules iteratively rather than atoms, our method further decouples the process from reactant sequence dependencies.

Building on this insight, we define an auxiliary retrosynthesis task, constraining the outputs of current retrosynthesis models to improve performance, and propose a RetroAux framework which is injected with molecular property knowledge as the prior. RetroAux first learns molecular chemical properties through multimodal knowledge of molecular-chemical text. Subsequently, RetroAux utilize diverse reactions to leverage chemical properties for retrosynthesis by learning latent correlations between product and reactants based on their properties. Departing from conventional methods, our method eliminates dependency on reaction templates or atom vocabularies. Insteadly, its prediction mechanism involves nearest-neighbor retrieval in a molecular embedding space, followed by inverse projection to generate reactants, as detailed in the lower part of Fig. 1. Finally, the framework further applies chemically grounded constraints to refine base model outputs. Empirical result shows RetroAux enhances **8** classical retrosynthesis methods, achieving consistent top-1 accuracy improvements averaging **2.22**%. Our core contributions include:

- To the best of our knowledge, we are the first to employ molecular property in retrosynthesis and find it effective.

- We introduce a new knowledge-driven methodology in retrosynthesis through auxiliary retrosynthesis task, enhancing prediction reliability through molecular property.

- Our plug-and-play knowledge-injection module enables seamless integration with all existing data-driven retrosynthesis models, achieving concistent performance improvements without retraining or requiring any architectural modifications.

## 2 RELATED WORK

### 2.1 MOLECULAR REPRESENTATION LEARNING

Current computational chemistry predominantly employs SMILES strings and molecular graphs for representation. SMILES strings (Weininger, 1988) compresses complex non-linear molecular structures into linear sequences, its syntax proves empirically difficult to learn with standard recursive architectures, requiring complex model designs and massive data to overcome the grammatical dependencies of linear representations (Wang et al., 2022), while graphs naturally preserve structural features (Li et al., 2023a). Molecular representation learning aims to map molecules to latent vectors that encode structural-property relationships. SMILES-based methods such as *SMILES-BERT* (Wang et al., 2019) leverage masked language modeling, while Graph-based approach *MolR* (Wang et al., 2022) preserve the reaction equivalence through GNNs.

### 2.2 RETROSYNTHESIS PREDICTION

Modern retrosynthesis methods fall into three paradigms. *Template-based* approaches (Chen & Jung, 2021) use predefined reaction templates to ensure chemical validity and interpretability, but their generalization is limited to known templates, failing on out-of-template reactions. *Semi-template-based* methods (Somnath et al., 2021) improve flexibility by identifying reaction centers and generating synthons, yet often rely on atom mappings during training, a ground-truth signal unavailable in real-world prediction, raising concerns about data leakage and overfitting. *Template-free* models (Han et al., 2024) treat retrosynthesis as a sequence generation task, offering strong generalization by predicting reactants autoregressively without explicit rules; however, they typically operate at the atom level and underutilize chemical knowledge. A detailed description of these approaches is provided in Appendix A.

## 3 METHOD

We propose RetroAux, consist of a Mol-Former and a Molecular Decoder, which leverages the fundamental principle that chemical reactions inherently depend on molecular properties. Read Appendix B.1 for more implementation details. The core idea of RetroAux is to constrain the retrosynthesis process by Molecular Decoder through chemically-informed embeddings from Mol-Former. Mol-Former maps molecules into vectors with explicit molecular chemical property semantics, so that the Molecular Decoder recursively generates reactants based on these vectors, then outputting corresponding molecule through *inverse projection*, maintaining molecular knowledge priors throughout the entire retrosynthesis workflow. The inference stage compares differences between our model's predictions and the base model's outputs to determine the final result. Fig. 2 shows the three stages of our method. Prior to detailing these modules, we formally define auxiliary retrosynthesis, as the methodological foundation.

### 3.1 PROBLEM STATEMENT

**Auxiliary Retrosynthesis.** We formly define the auxiliary retrosynthesis task, while the retrosynthesis task involves inferring all potential reactants from a given product, the auxiliary retrosynthesis task predicts the remaining reactants using both the product and the first reactant predicted by the base model, outputting more reliable predictions to enhance the base model's performance. Given product $P$ and the first predicted reactant $R_1$ from an base model, the auxiliary task aims to predict remaining reactants $\{R_2, ..., R_n\}$ through:

$$\{R_2, ..., R_n\} = \arg\max_{\mathcal{R}} \Phi(P, R_1, \mathcal{R}|\Theta), \tag{1}$$

where $\Phi(\cdot)$ denotes model parameterized by $\Theta$, $\mathcal{R}$ denotes solution retrosynthesis prediction space.

### 3.2 CROSS-MODAL MOLECULAR CHEMICAL KNOWLEDGE LEARNING

**Injecting Chemical Knowledge via Multimodal Pretraining.** RetroAux leverages **multimodal pretraining** for enhanced robustness. Unlike SMILES-only methods, our framework is pretrained

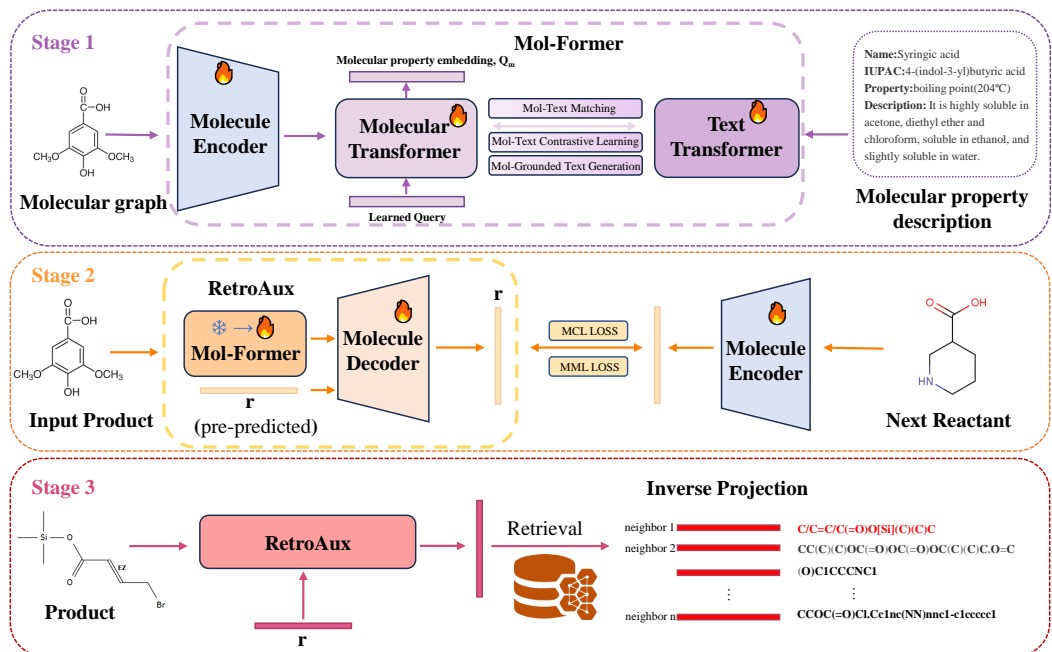

Figure 2: **RetroAux involves three separated stages. Stage 1:** Mol-Former encodes molecules into molecular property embeddings $\mathbf{Q}_m$, trained on molecular-chemical text multimodal data. **Stage 2:** Molecular Decoder is trained through reformulated objective (3.3): it takes $(Q_p, r_1, r_2, \ldots, r_{k-1})$ as inputs to predict $(r_1, r_2, \ldots, r_K)$. **Stage 3:** Output molecular through Inverse Projection.

on large-scale chemical-text datasets, equipping it with broader chemical intuition to recognize and appropriately respond to phenomena like activity cliffs.

We introduce **Mol-Former**, a multimodal encoder that aligns molecular and textual representations through a shared embedding space. It comprises a Molecular Transformer, a Text Transformer, and a Molecular Encoder. The architecture follows the Q-Former design (Li et al., 2023b), where a learnable Molecular Transformer attends to facilitate interaction between molecular representations and textual chemical knowledge, ultimately transforming the query tokens into molecular property embeddings $\mathbf{Q}_m$ with explicit chemical semantics. Formally, given a SMILES string and its associated text description, we first encode them into latent sequences using pretrained molecular encoder MolR (Wang et al., 2022) and a Text Transformer. A learnable query then attends to both modalities via cross-attention, yielding a joint embedding space. To align molecular structures with their semantic descriptions, we train Mol-Former using a contrastive and matching objective over the paired data. Specifically, we optimize:

$$\mathcal{L} = \mathcal{L}_{\text{MTC}} + \mathcal{L}_{\text{MTM}} + \mathcal{L}_{\text{LM}}, \tag{2}$$

where $\mathcal{L}_{\text{MTC}}$ is a molecular-text contrastive loss encouraging similar embeddings for matched pairs, $\mathcal{L}_{\text{MTM}}$ promotes accurate identification of positive pairs, and $\mathcal{L}_{\text{LM}}$ enables generation of chemically meaningful descriptions conditioned on the molecule.

This cross-modal alignment objective forces the model to ground textual semantics, such as "carboxylic acid", "R-enantiomer", or "Ketone", into the structure-derived molecular representation, equipping it with broader chemical intuition to recognize phenomena like activity cliffs. Ultimately Mol-Former transforms the query tokens into molecular property embeddings $\mathbf{Q}_m$ with explicit chemical semantics:

$$\mathbf{Q}_m = \text{Mol-Former}(\mathbf{m}), \tag{3}$$

where $\mathbf{m}$ denotes molecular embeddings generated by molecular encoder. This integration of chemical knowledge into $\mathbf{Q}_m$ enables subsequent utilization of molecular chemical properties as *a prior* guidance for retrosynthesis prediction.

### 3.3 MOLECULAR-LEVEL RETROSYNTHESIS PREDICTION VIA AUTOREGRESSIVE MOLECULAR DECODER

**Molecular-Level Autoregressive Prediction.** To better capture the compositional nature of chemical reactions, we formulate retrosynthesis as an autoregressive sequence generation task at the molecular level, where each step generates a complete reactant molecule rather than individual atoms, promoting higher-level chemical reasoning. We define the solution retrosynthesis prediction space $\mathcal{R}$ as a sequence of molecular mappings $(m_1, \ldots, m_K)$, each representing a reactant in the context of the product. Given the product embedding $\mathbf{Q}_{pro}$ from Mol-Former and previous predictions, the Molecular Decoder models the conditional likelihood:

$$P(\mathcal{R}) = \prod_{k=1}^{K} P(m_k \mid m_{i<k}, \mathbf{Q}_{pro}), \tag{4}$$

where $m_k$ denotes the $k$-th reactant. During training, we ground predictions on ground-truth reactants from standard datasets to avoid potential bias or performance limitations that might arise from relying on any base model's predictions, which enables universal compatibility across diverse predictors. We employ teacher forcing (Feng et al., 2021), conditioning each step on true preceding reactants.

**Learning Discriminative Reactant Representations.** To ensure the Molecular Decoder generates both accurate and chemically meaningful reactants, we design a dual-objective training scheme that enforces precise prediction and enhances discrimination between structurally similar molecules.

First, a **M**olecular **C**ontrastive **L**oss minimizes the L2 distance between predicted reactant embeddings $\mathbf{r}$ and their targets $\mathbf{m}$:

$$\mathcal{L}_{\text{MCL}} = \frac{1}{N} \sum_{i=1}^{N} \|\mathbf{r}_i - \mathbf{m}_i\|_2^2. \tag{5}$$

ensuring precise reconstruction of known reactants. Second, a **M**olecular **M**atching **L**oss incorporates hard negative sampling to distinguish structurally similar but functionally distinct molecules:

$$\mathcal{L}_{\text{MML}} = -\log \frac{\exp(\mathbf{r}_i^\top \mathbf{m}_i / \tau)}{\sum_{j \in \mathcal{N}_i} \exp(\mathbf{r}_i^\top \mathbf{m}_j / \tau)}, \tag{6}$$

where $\mathcal{N}_i$ contains hard negative samples and $\tau$ is a temperature parameter.

Training proceeds in two phases: (1) Mol-Former is frozen while the Molecular Decoder focuses purely on learning correlations between product and reactant properties without interference from encoder updates. (2) end-to-end fine-tuning refines both Mol-Former and Molecular Decoder. The overall objective is:

$$\mathcal{L}_{\text{Decoder}} = \mathcal{L}_{\text{MCL}} + \lambda \mathcal{L}_{\text{MML}}, \tag{7}$$

guiding the model to precisely learn the chemical property relationships between the product and its corresponding reactants. Furthermore, our property-based filtering approach is fundamentally unbiased toward rare but valid reaction pathways. Unlike purely data-driven methods, which often struggle to learn underrepresented pathways due to limited training examples, our framework evaluates reactions based on reactants' intrinsic chemical properties rather than statistical prevalence. This ensures equitable consideration of all chemically plausible pathways, regardless of their frequency, making RetroAux does not ignore valid but rare reaction pathways.

### 3.4 RETRIEVING MOLECULES FROM A DYNAMIC MOLECULAR DICTIONARY

The reconstruction of SMILES strings from vector representations $\mathbf{r}$ poses non-trivial challenges. First, the solution retrosynthesis prediction space $\mathcal{R}$ and the ground-truth molecular embedding space are inherently non-equivalent, as they originate from distinct model outputs. Second, $\mathbf{r}$ predicted by the autoregressive Molecular Decoder cannot maintain precise one-to-one correspondence with actual molecules. To resolve this issue and enable deterministic molecular output in SMILES format, we introduce Molecular Dictionary.

**Molecular Dictionary.** The molecular dictionary serves as a fundamental key-value store, wherein the values correspond to molecular embeddings, and the keys are the associated molecular SMILES strings. For benchmarking baseline performance on standard datasets, the initial dictionary is

populated with molecules drawn exclusively from the dataset. However, real-world chemical synthesis is not confined to the boundaries of any fixed dataset. Consequently, the molecular dictionary is designed to be dynamic rather than static. This dynamic dictionary automatically incorporates candidate molecules predicted by the base model into its repository. This mechanism enables RetroAux to continuously adapt to novel chemical spaces and discover new molecules absent from the original dictionary. In practical deployment scenarios, the molecular dictionary can also be manually augmented in bulk with catalogs of molecules from real-world chemical vendors (e.g., Enamine, SciFinder). The efficacy and computational efficiency of this dynamic dictionary strategy are analyzed in Sec 4.5.

Formally, given a predicted embedding $\mathbf{r}$, we retrieve the most chemically similar molecule $M_p$ from a dictionary $\mathcal{D}$ of known compounds:

$$M_p = \arg\min_{M \in \mathcal{D}} \|\mathbf{r} - \mathbf{m}_M\|, \tag{8}$$

where $\mathbf{m}_M$ is the embedding of molecule $M$, computed using the same molecular encoder (e.g., MolR). This nearest-neighbor search ensures that the output molecule is both valid and structurally compatible with the predicted chemical properties.

### 3.5 INFERENCE STRATEGY FOR AUXILIARY RETROSYNTHESIS

In single-step retrosynthesis the set of reactants corresponding to a given product is inherently unordered. Our analysis however reveals that the L2 distances between the embedding of each reactant and the embedding of the product are not uniform. A smaller L2 distance indicates greater chemical similarity between the reactant and the product. This observation is consistent with chemical intuition which suggests that reactants exhibit varying degrees of similarity to the product.

To leverage this property without imposing artificial orderings, we adopt the following inference strategy: (1) Rank candidate reactants from the base model by their L2 distance to the product embedding. (2) Initialize autoregressive generation using both the closest (most similar) and farthest (least similar) reactants as starting points. (3) Aggregate the inference results obtained from these two distinct sampling strategies. (4) Aggregate results via intersection with the original prediction. This procedure is detailed in Algorithm. Algorithm details are in the Appendix B.2. By operating post-hoc on the base model's outputs, RetroAux constrains predictions without modifying the base model's architecture, ensuring seamless integration and broad compatibility across frameworks.

## 4 EMPIRICAL RESULTS

### 4.1 SETUP

**Datasets.** We pretrain Mol-Former on three chemical-text multimodal datasets: ChEBI-20-MM (Edwards et al., 2021) (33K pairs), Mol-Instructions (Fang et al., 2024) (734K selected entries), and PubChemSTM (Liu et al., 2023) (280K+ pairs). Our analysis shows that **20.81**% of the data explicitly describes critical chemical attributes, such as functional groups and chirality, **70.13**% cover broader chemical features. Please refer to Appendix C.1 for more details. For evaluation, we use USPTO-50K (Schneider et al., 2016), a widely-used benchmark. We adopt the data split of Coley et al. (2017b), canonicalize SMILES using RDKit (Landrum et al., 2013), remove atom mappings to prevent leakage, and exclude single-reactant reactions because they often imply implausible direct transformations that are unlikely to occur without additional reagents or catalysts in real-world synthesis.

**Evaluation Metrics and Baselines.** Following prior work, we report top-1 accuracy based on exact match of canonical SMILES between prediction and ground truth, without using atom mappings or reaction class labels—reflecting real-world deployment conditions. We evaluate eight representative methods across paradigms: *Template-based*: LocalRetro (Chen & Jung, 2021); *Semi-template*: GraphRetro (Somnath et al., 2021); *Template-free*: RetroWise (Zhang et al., 2024), EditRetro (Han et al., 2024), NAG2G (Yao et al., 2024), R-SMILES (Zhong et al., 2022), Retroformer (Yao et al., 2022), and Transformer (Vaswani et al., 2017). To ensure fair comparison, all baseline results are re-evaluated using publicly released model weights.

Table 1: **Without** and **with** pre-trained Molecular Decoder Top-1 accuracy on USPTO-50K Dataset. $\Delta$ indicates the improvement after using RetroAux.

| Method | Origin Acc. | w/o Pre-trained Decoder | | w/ Pre-trained Decoder | |
|---|---|---|---|---|---|
| | | +Ours | $\Delta$ | +Ours | $\Delta$ |
| *Template-based* | | | | | |
| LocalRetro (Chen & Jung, 2021) | 52.58 | 54.60 | **2.02** | 55.34 | **2.76** |
| *Semi-template-based* | | | | | |
| GraphRetro (Somnath et al., 2021) | 53.72 | 55.98 | **2.26** | 56.64 | **2.92** |
| *Template-free* | | | | | |
| Transformer (Vaswani et al., 2017) | 42.80 | 47.87 | **5.07** | 48.67 | **5.87** |
| Retroformer (Yao et al., 2022) | 52.89 | 55.46 | **2.57** | 56.28 | **3.39** |
| R-SMILES (Zhong et al., 2022) | 53.06 | 53.90 | **0.84** | 55.46 | **1.40** |
| NAG2G (Shi et al., 2020; Yao et al., 2024) | 54.66 | 56.88 | **2.22** | 57.70 | **3.04** |
| EditRetro (Han et al., 2024) | 60.33 | 62.07 | **1.74** | 62.89 | **2.56** |
| RetroWise (Zhang et al., 2024) | 64.71 | 65.73 | **1.02** | 66.33 | **1.62** |

Table 2: Ablation Study of Modules and Objectives on USPTO-50K.

| Settings | Modules | | Objectives | |
|---|---|---|---|---|
| | MF | MML | MCL | Top-1 (%) |
| *Module Ablation* | | | | |
| (a) | – | ✓ | ✓ | 94.74 |
| *Objective Ablation* | | | | |
| (b) | ✓ | ✓ | – | 92.86 |
| (c) | ✓ | – | ✓ | 94.86 |
| (d) | ✓ | ✓ | ✓ | **96.15** |

Table 3: Ablation Study of Molecular Encoder and Text Input.

| Settings | Modules | | Objectives | |
|---|---|---|---|---|
| | MolR | Bbs | Text | Top-1 (%) |
| *GNN Model* | | | | |
| (a) | – | ✓ | – | 90.44 |
| (b) | – | ✓ | ✓ | 92.87 |
| *Seq Model* | | | | |
| (c) | ✓ | – | – | 92.97 |
| (d) | ✓ | – | ✓ | **94.01** |

## 4.2 MAIN RESULT

**Results on USPTO-50K.** RetroAux enhances accuracy across all evaluated models on USPTO-50K (Schneider et al., 2016), including all three categories of single-step retrosynthesis methods, where $\Delta$ in the Tab. 1 denotes improvement margins. Notably, the average **2.22%** improvement is achieved through "Zero-shot" integration, without any architectural modifications or retraining of base models. We observe over 2% improvements even for template-based and semi-template-based models which are considered limited in generalization ability previously. The result demonstrates the universal applicability and robust capability of RetroAux.

**Transfer Learning.** We further investigate RetroAux's transfer learning capability by pre-training its Molecular Decoder on 143K reactions from Mol-Instructions (Loshchilov & Hutter, 2019) and fine-tuning on USPTO-50K. As shown in Tab. 1, scaling up the training data enhances RetroAux's comprehension of molecular property interactions in chemical reactions, increasing the average improvement margin from **2.22**% to **2.95**%. This demonstrates RetroAux's capacity to leverage expanded chemical knowledge for stronger performance gains while maintaining parameter efficiency.

## 4.3 ABLATION STUDY

We conduct ablation studies to investigate contributions from different model components. First, we evaluate the impact of Mol-Former. As shown in Tab. 2, Setting a directly connects the molecular encoder to the Molecular Decoder, while Setting b incorporates Mol-Former. By using the Mol-Former ,the Top-1 accuracy increases **1.41%** on the auxiliary retrosynthesis task, demonstrating the strength of molecular chemical knowledge injection. Additionally, we further assess the roles of individual loss functions in the Molecular Decoder. Each single loss function achieves reasonably high accuracy, while their combined usage yields optimal performance. This indicates the simplicity

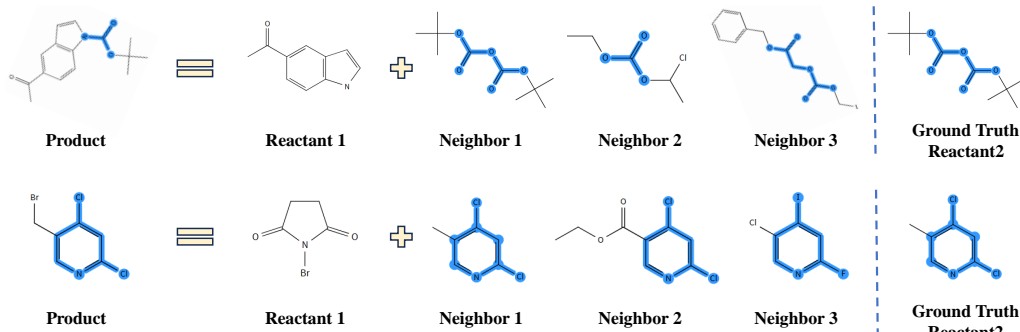

Figure 3: **Case study of retrieved molecules.** The same molecular structure is highlighted by blue background. The first column shows the ground truth, the penultimate column and the last column show the products and the first reactant, and the rest show three neighbourhood targets.

yet effectiveness of our training objectives. The term *top-1 accuracy* here refers to the proportion of correct predictions for the next reactant in the auxiliary retrosynthesis task, where the model is given a product molecule and one reactant to predict the remaining reactant(s).

To investigate the contribution of textual chemical knowledge to molecular representation robustness, we conduct ablation studies on the USPTO-50K dataset. The term *top-1 accuracy* reported here is as same as the evaluation setting in Section 4.3. Text properties significantly enhance molecular representation robustness. Our experiments confirm that incorporating chemical multimodal text consistently improves performance across tasks, demonstrating the critical role of textual knowledge in enhancing molecular understanding. Tab. 3 is the comparison. The choice of molecular encoder impacts performance, but text integration remains beneficial regardless. When replacing MolR (GNN-based) with Bert-base-smiles, a bidirectional Transformer pretrained on SMILES strings, the model still achieves competitive results in retrosynthesis tasks. The slightly lower performance of Bert-base-smiles may stem from the inherent advantage of graph-based representations for capturing molecular structural features. All experiments were conducted within 100 epochs with early stopping due to computational constraints.

## 4.4 CASE STUDY

We conduct case studies on USPTO-50K by randomly selecting two reactions involving two reactants for intuitive analysis to better understand whether the model can correctly utilize molecular properties in chemical reactions, with results shown in Fig. 3. RetroAux retrieves three candidate nearest-neighbor molecules for the second reactant through inverse projection. The first example (top panel of Fig. 3) demonstrates the synthesis of tert-butyl 5-acetylindole-1-carboxylate. Neighbor 1 matches the ground truth, corresponding to a Boc protection reaction (nucleophilic acyl substitution reaction) of the amino group. The blue highlights indicate that other neighbors share the same carbonate ester functional group with ground truth, which could also serve as reactive centers for nucleophilic acyl substitution. The product exhibits a clear N-Boc derivative structure. (See Appendix C.2 for further embedding visualization.) The second example (bottom panel of Fig. 3) illustrates a radical bromination reaction (halogenation reaction) where the product 5-(bromomethyl)-2,4-dichloropyridine retains the structural features of the second reactant NBS – a characteristic of this reaction type. Other phenomena are similar to the first example.

These examples demonstrate that through molecular property injection, our method inherently learns fundamental reaction rules, genuinely comprehends complex property correlations between products and reactants, and effectively leverages molecular properties for retrosynthesis prediction. Notably, even wrong reactants predicted by RetroAux retain chemical relevance, enabling chemists to infer potential reaction types through analysis of molecular property patterns in the predictions.

Figure 4: **Experimentally Validated Novel Reaction Pathways Predicted by RetroAux Using Dynamic Molecular Dictionary.** RetroAux, leveraging its dynamic molecular dictionary, predicted three previously unreported synthetic routes including Suzuki-Miyaura coupling, Bucherer reaction, and Friedel-Crafts acylation, involving molecules entirely absent from its initial dictionary.

## 4.5 Effectiveness and Efficiency of the Dynamic Molecular Dictionary

To further validate the effectiveness of the dynamic molecular dictionary, we conducted wet-lab experiments to assess its capacity for novel molecule discovery and generalization, supplementing the standard benchmark evaluations presented in Sec 4.2. Specifically, leveraging the dynamic dictionary, RetroAux successfully predicted **three entirely novel reaction pathways**: a Suzuki-Miyaura coupling, a Bucherer reaction, and a Friedel-Crafts acylation, as illustrated in Fig 4. Crucially, all molecules involved in these newly predicted pathways were absent from the initial molecular dictionary prior to prediction. The successful experimental validation of these predictions provides compelling empirical evidence that the model, when based on the dynamic dictionary, possesses exceptional generalization capabilities for discovering both novel molecules and viable synthetic routes.Our analysis further shows that the retrieval process is both effective and efficient. Using a top-1 nearest neighbor search, the dictionary provides sufficient chemical constraints to guide accurate predictions, with diminishing returns observed for larger $K$ values. Despite this, the computational cost remains negligible. KNN search accounts for less than 1% of total inference time even with a 200K-molecule dictionary. For details on the ablation study of retrieval size ($K$) and computational efficiency across different dictionary scales, please refer to Appendix C.3.

## 5 Limitations and future work

In this work, we initiate the first exploration of knowledge-driven methodologies in retrosynthesis. RetroAux is designed as an auxiliary model that maintains output length consistency with the base model and requires at least one reactant as the starting point for reaction pathway prediction. Thus, it does not operate for single-reactant reactions. This design choice stems from its core objective: to learn reactant-product relationships rather than independently inferring pathways from products alone. The model learns the reaction direction based on a given product and a reactant, capturing the "difference" between their embeddings(Appendix C.2 provides a more detailed discussion). This inherently requires at least one reactant input to effectively model the relationship. Nevertheless, even in scenarios involving single-reactant reactions, RetroAux still improves the base model's prediction accuracy by an average of 2.22%, demonstrating its value as a plug-and-play auxiliary framework.

We point out two future directions: First, while our primary focus is improving single-step retrosynthesis, RetroAux can be iteratively applied to break down multi-step problems into sequential single-step predictions. Indeed, single-step retrosynthesis prediction serves as a critical component in generating search options for multi-step retrosynthesis planning, as seen in approaches such as Retro*, RetroGraph and PDVN. Our method enhances this process by providing more plausible reaction nodes during the construction of the retrosynthesis tree. Second, as noted in Section 3.4, resolving the molecular uniqueness problem, which entails determining unique molecular structures from continuous embeddings, remains crucial given the infinite molecular space.

## 6 Conclusion

We propose the auxiliary retrosynthesis task and introduce RetroAux, a model that leverages molecular property knowledge to enhance existing retrosynthesis approaches. We demonstrate that systematic constraints through molecular property knowledge can consistently improve the performance of current retrosynthesis models. Visualization results further substantiate the critical role of molecular properties in retrosynthetic planning.

## 7 ETHICS STATEMENT

This work introduces RetroAux, a knowledge-driven framework for enhancing single-step retrosynthesis prediction through molecular chemical knowledge injection. Our research does not involve human subjects, animal testing, or sensitive personal data. All experiments are carried out on publicly available chemical reaction datasets (USPTO-50K (Schneider et al., 2016), ChEBI-20-MM (Edwards et al., 2021), Mol-Instructions (Fang et al., 2024), and PubChemSTM (Liu et al., 2023)), used in accordance with standard academic research licenses and chemical data usage norms. We do not release any proprietary or confidential molecular data, nor do we generate hazardous or restricted chemical structures. The goal of this work is not to enable malicious synthesis but to improve the reliability and chemical plausibility of AI-assisted retrosynthetic planning, thereby supporting safer and more efficient drug discovery and organic synthesis. By injecting established chemical knowledge, RetroAux aims to align AI predictions with scientific principles, reducing the risk of chemically invalid or unsafe suggestions. We believe that this knowledge-driven approach promotes responsible AI for science and aligns with the ethical imperative to advance public good through trustworthy, interpretable, and scientifically grounded machine learning.

## 8 REPRODUCIBILITY STATEMENT

We have taken comprehensive steps to ensure the reproducibility of our work. All implementation details including model architecture, training objectives, hyperparameters and optimizer settings, which are fully described in Sec. 3 and Sec. 4 of the main paper. The evaluation protocol, dataset splits and SMILES canonicalization method (using RDKit) follow established conventions and are explicitly documented. Additional technical details, such as embedding visualizations and chemical analysis, are provided in the Appendix C. In the supplementary material, we provide additional implementation notes and pseudo-code for guidance. Together, these resources are intended to make it straightforward for researchers to reproduce and verify our findings.

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

## A  MORE RELATED WORKS

**Template-Based Methods.** These approaches utilize reaction templates—either manually curated (Hartenfeller et al., 2011; Szymkuć et al., 2016) or algorithmically extracted (Coley et al., 2017a; Designer, 2009)—to encode reaction center patterns. Formulating retrosynthesis as template retrieval, representative works include: *RetroSim* (Coley et al., 2017b) using molecular fingerprint similarity; *NeuralSym* (Segler & Waller, 2017) with template classification by deep neural network; *GLN* (Dai et al., 2019) employing graph neural networks for joint template-reactant modeling; and state-of-the-art *LocalRetro* (Chen & Jung, 2021) combining local reaction center learning with global attention. While providing chemical interpretability, such methods fundamentally limit generalizability to out-of-template reactions.

**Semi-Template-Based Methods.** Semi-template approaches follow a two-stage workflow: 1) Identify reaction centers via atom mapping and split products into synthons; 2) Convert synthons into complete reactants. *RetroXpert* (Yan et al., 2020) implements this through graph attention mechanisms. *G2Gs* (Shi et al., 2020) uses GNNs to estimate reactivity scores for atom pairs to designate the reaction center. *GraphRetro* (Somnath et al., 2021) predicts graph edits to obtain synthons before attaching leaving groups. *NAG2G* (Yao et al., 2024) achieves precise atom mapping via node alignment. However, Atom mapping provides explicit reactant-product atom correspondence, which is unknown in real retrosynthesis, since the task of retrosynthesis itself is to predict reactants, it is impossible to know in advance the atomic mapping relationship between the product and the reactants at the time of input, making such mappings potential sources of data leakage (Maziarz et al., 2025).

**Template-Free Methods.** Most template-free methods reframe retrosynthesis as a sequence-to-sequence translation task between product and reactant SMILES strings. Autoregressive methods dominate template-free single-step retrosynthesis. Recent state-of-the-art (SOTA) approaches universally adopt this paradigm, as it iteratively generates atoms to form reactant sequences without imposing order constraints, only the correctness of the final reactant set matters. Karpov et al. (Karpov et al., 2019) pioneered Transformer-based models (Vaswani et al., 2017) for this task. Building on this, *SCROP* (Zheng et al., 2019) integrated syntax correction modules to resolve invalid SMILES generation. *Retroformer* (Yao et al., 2022) enhanced information exchange between local reactive regions and global contexts through local attention heads. *R-SMILES* (Zhong et al., 2022) aligned product-reactant SMILES pairs to minimize edit distances. *EditRetro* (Han et al., 2024) improved prediction accuracy and diversity through string editing strategies. *PMSR* (Jiang et al., 2023) achieved state-of-the-art performance via three pretraining tasks that capture retrosynthetic chemical rules. While template-free methods exhibit strong generalization, they primarily generate from an atom perspective with insufficient utilization of chemical knowledge.

## B  IMPLEMENTATION AND INFERENCE DETAILS

### B.1  IMPLEMENTATION DETAILS

As aforementioned, we adopt the original Q-Former architecture and MolR (Wang et al., 2022) framework to implement Mol-Former. MolR (Wang et al., 2022) is initialized with weights from its official GitHub repository. We employ a minimalist design, a standard transformer decoder (Vaswani et al., 2017) with 6 layers and 8 attention heads as our Molecular Decoder, but remove the final softmax layer to enable direct vector output. We set 192 as the maximum text token length and use AdamW optimizer (Loshchilov & Hutter, 2019) with the peak learning rate $1 \times 10^{-3}$. The pretraining process completed 351K steps on 4 RTX4090 GPUs, then we train Molecular Decoder on USPTO-50K with reaction type unknown for 200 epochs using the AdamW optimizer with a learning rate of 1e-4. The evaluation in Sec.4.2 demonstrates that this simple architecture, when combined with our inverse projection strategy, significantly enhances performance of base model without requiring complex architectural modifications.

### B.2  INFERENCE ALGORITHM

Algorithm 1 details the RetroAux inference workflow. Given a product SMILES $m_p$, it first retrieves an initial reactant set $R_o$ from the base model and embeds molecules via Mol-Former. It then identifies the closest ($r_c$) and farthest ($r_f$) reactants in embedding space, generates auxiliary edits

---

**Algorithm 1** RetroAux Inference Workflow

---

**Input** Product SMILES $m_p$
**Output** Final reactant set $R_{\text{final}}$
  1: $R_o \leftarrow \text{BaseModel}(m_p)$
  2: $f(\cdot) \leftarrow \text{Mol-Former}(\cdot)$
  3: $r_c \leftarrow \arg\min_{r \in R_o} \|f(r) - f(m_p)\|_2$             # Closest prediction
  4: $r_f \leftarrow \arg\max_{r \in R_o} \|f(r) - f(m_p)\|_2$            # Farthest prediction
  5: $E_c \leftarrow \text{RetroAux}(m_p, f(r_c))$           # Auxiliary edits from closest
  6: $E_f \leftarrow \text{RetroAux}(m_p, f(r_f))$         # Auxiliary edits from farthest
  7: $R_c \leftarrow \{r_c\} \cup P^{-1}(E_c)$            # Reactants from closest path
  8: $R_f \leftarrow \{r_f\} \cup P^{-1}(E_f)$           # Reactants from farthest path
  9: $R_{\text{candidates}} \leftarrow R_c \cup R_f$
10: $R_{\text{intersect}} \leftarrow R_{\text{candidates}} \cap R_o$
11: **if** $R_{\text{intersect}} \neq \emptyset$ **then**
12:     $R_{\text{final}} \leftarrow R_{\text{intersect}}$
13: **else**
14:     $R_{\text{final}} \leftarrow R_c$           # Fallback to closest-augmented set
15: **end if**
16: **return** $R_{\text{final}}$

---

from both, and constructs candidate reactant sets $R_c$ and $R_f$. The final prediction is obtained by intersecting these candidates with the original base model output $R_o$; if the intersection is empty, it falls back to $R_c$. This post-hoc refinement preserves compatibility while enhancing prediction reliability without modifying the base model.

# C   Supplementary Analysis and Results

## C.1   Datasets

In selecting the dataset, we prioritized chemically relevant multimodal datasets that comprehensively capture key reaction properties. Our analysis shows that 20.81% of the data explicitly describes critical chemical attributes, such as functional groups and chirality. The remaining 70.13% of the data detail other molecular properties, such as electronic states and biochemical roles. See Fig. 5 for details.

## C.2   Embedding visualization

Further embedding visualization provides deeper insight into how RetroAux leverages molecular property space. As shown in Fig. 7, the predicted vector for the second reactant in the Boc protection example precisely resides within the *carbonate ester* functional group cluster and is oriented more closely towards *hydrocarbon* directions in the embedding space. This geometric alignment indicates that RetroAux successfully captures the alkane substructure present in the product molecule and correctly attributes it to the second reactant, demonstrating its ability to perform chemically grounded reasoning through property-aware representation learning.

Additionally, we illustrate the importance of molecular properties in chemical reactions through ketone reduction and alcohol reduction examples. Their reaction templates are R-CHO + $H_2 \rightarrow RCH_2OH$ and $RCH_2OH \xrightarrow{\text{HI, Zn/HCl}} RCH_3$ respectively. We first generate molecular property embeddings for Propan-2-one ($CH_3COCH_3$) and 2,3-butanedione ($CH_3COCOCH_3$) using the pretrained Mol-Former model, along with corresponding alcohols and hydrocarbons. These vectors are then visualized via t-SNE (Van der Maaten & Hinton, 2008), as shown in Fig. 6. The results clearly demonstrate: (1) Molecules with similar SMILES but different properties form distinct clusters in Mol-Former's chemical space. This provides embedding-level molecular property priors during retrosynthesis prediction, enabling chemically grounded reasoning, a departure from conventional approaches relying solely on latent pattern learning from massive reaction data. (2) Homologous reactions exhibit analogous directional patterns in the projected low-dimensional property space. This facilitates

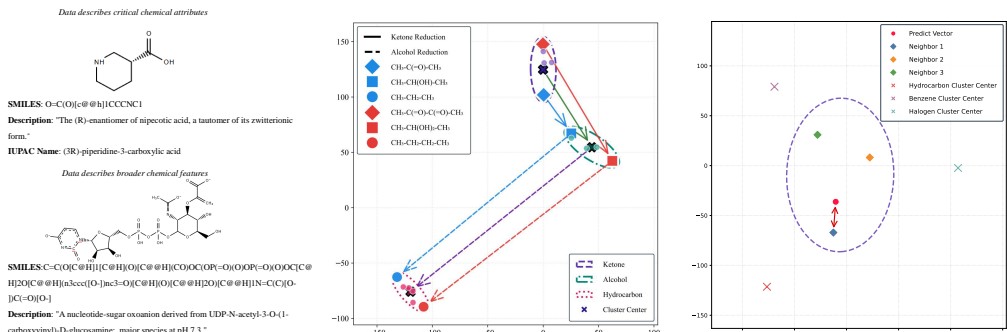

Figure 5: Molecular Property Annotation in Pretraining Data.

Figure 6: Reaction visualization of ketone reduction and alcohol reduction.

Figure 7: Embedding visualization of predicted vector by RetroAux.

learning common characteristics and inter-reaction correlations. The model appears to capture functional group quantity effects, as evidenced by the red dashed arrow (representing dual ketone group reduction in 2,3-butanedione) being approximately twice as long as the blue arrow (single ketone reduction).These geometric regularities in chemical space enable more interpretable and chemically plausible prediction compared to traditional black-box approaches.

### C.3 RETRIEVED SIZE AND COMPUTATIONAL EFFICIENCY

Tab. 4 demonstrates how retrieval size (the number of neighbors) affects constraint effectiveness. Specifically, we evaluate $K \in \{1, 3, 5, 10, 50\}$ during nearest-neighbor search and report top-1 accuracy when applied to EditRetro.

Table 4: Study on the number of neighbors.

| Neighbors | 1 | 3 | 5 | 10 | 50 |
|---|---|---|---|---|---|
| Accuracy | 62.07 | 63.11 | 63.21 | 63.33 | 63.49 |

The top-K accuracy aligns with standard retrieval-based evaluation in reaction prediction (RetroKNN[1]), where a prediction is considered correct if any of the top-K retrieved candidates matches the ground truth, rather than post-retrieval ranking. From these results, we first observe that incorporating a single retrieval ($K = 1$) improves accuracy from 60.33% to 62.07%. When $K \geq 5$, accuracy further increases to approximately 63.21%, with no significant improvements observed beyond this threshold.Increasing $K$ to larger values neither substantially boosts performance nor causes notable degradation. We hypothesize this occurs because sufficient constraint information is already captured at $K = 5$ to guide the base model's predictions, while molecules farther from the query provide diminishing marginal utility due to their weaker chemical relevance to the target reaction.

In addition to validating its effectiveness, we also conducted a systematic investigation into the computational efficiency of querying the dynamic molecular dictionary. As shown in Tab. 5, we measured the kNN search time across varying molecular dictionary sizes on an RTX 4090 GPU, topK=50, and batch size=32.

Table 5: Computational efficiency.

| Vocabulary Size | 10,000 | 100,000 | 200,000 |
|---|---|---|---|
| Latency (ms) | 0.51 | 8.38 | 15.01 |

KNN search accounts for less than 1% of total inference time even with a 200K vocabulary size (15.01ms / 2.19s). This demonstrates that the retrieval step introduces negligible latency compared to the full inference pipeline. For large-scale deployment, we recommend using the Milvus database for storing and searching molecular vectors. Milvus is a highly optimized vector database system, benchmarks show search latencies below 0.3s even for 50M vectors.

### C.4 EXPERIMENTALLY CHEMICAL ANALYSIS

To verify the generalization ability of our prediction model in real-world scenarios, we conducted experiments in a chemistry laboratory based on the model's predictions. In chemistry, the identification of a chemical substance is typically achieved through the analysis of its spectra. The products

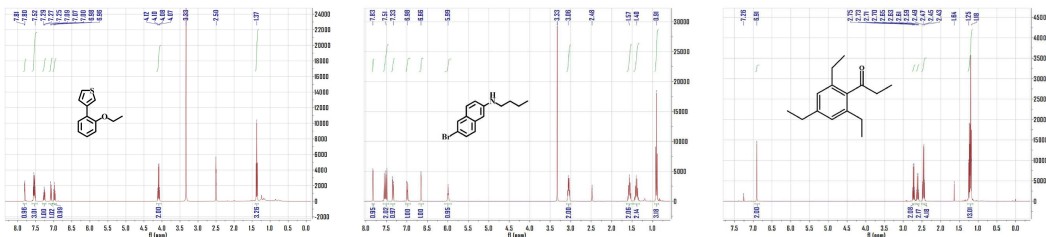

Figure 8: ¹H NMR spectra of all chemical products

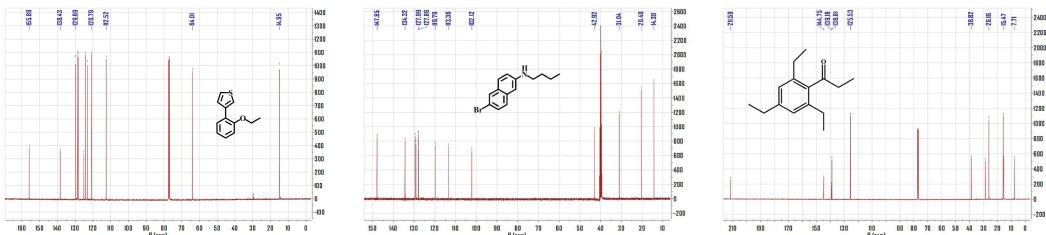

Figure 9: ¹³C NMR spectra of all chemical products

were sent to a third-party testing institution for analysis, where nuclear magnetic resonance (NMR) spectroscopy was used to determine whether the reaction products matched the predictions.As shown in Fig. 8 and Fig. 9, the following images show the hydrogen (¹H NMR) and carbon (¹³C NMR) spectra of the predicted reaction product, which were found to be in complete agreement with the model's predictions after analysis. The order from left to right is Suzuki–Miyaura coupling, Bucherer reaction, and Friedel–Crafts acylation.

## D    STATEMENT

LLMs were used for grammar checking. No substantive edits requiring disclosure.

