# OpenReview forum: "RetroAux: Boosting Retrosynthesis via Molecular Property-Aware Learning"
_ICLR.cc/2026/Conference — ICLR 2026 Conference Withdrawn Submission_

### Official Review · Reviewer_BqNf · 2025-10-21

**Soundness:** 2
**Presentation:** 2
**Contribution:** 2
**Rating:** 2
**Confidence:** 4

**Summary:**

The authors provide RetroAux, a post-hoc method to improve the existing base models performance on retrosynthesis task.

**Strengths:**

- From Table 1, it seems that the method is general applicable to many existing retrosynthesis systems.
- The retrieval based method more explainable.

**Weaknesses:**

Major:

-	The authors claim that “we are the first to employ molecular property in retrosynthesis and find it effective”. Yet, many chemical/scientific foundations models are also pre-trained with rich molecular properties & knowledge. These models, such as NatureLM [1], are also effective in retro synthesis tasks.

-	How do authors control data leakage? e.g., the training data on Sec 4.1 may already contain information of USPTO50k test set products, or even worse the exact or similar reactions, by text or SMILES.

-	The method is only evaluated in USPTO50k, not larger datasets such as USPTO-MIT or USPTO-FULL.

-	The authors are encouraged to discuss and compare with other retrieval-based retrosynthesis models like RetroKNN[2]

Minor:

-	The line 299-300, i.e., “Aggregate results via intersection with the original prediction. This procedure is detailed in Algorithm. Algorithm details are in the Appendix B.2.” should be revised. Which ‘algorithm’ is mentioned in this sentence?

Refs:

-	[1] Nature Language Model: Deciphering the Language of Nature for Scientific Discovery

-	[2] Retrosynthesis Prediction with Local Template Retrieval

**Questions:**

- Some reactions only contains single reactant. Are RetroAux still useful in this case?

---

### Official Review · Reviewer_nMds · 2025-10-26

**Soundness:** 2
**Presentation:** 2
**Contribution:** 2
**Rating:** 2
**Confidence:** 5

**Summary:**

This paper proposes a three-stage framework that injects chemical priors into retrosynthesis prediction. In Stage 1 (pre-training), a multimodal model ingests molecular structures and associated text, optimized with (i) a molecule–text contrastive loss, (ii) a boosted-positive objective, and (iii) a chemical-description prediction loss. Stage 2 switches to autoregressive molecule prediction. Stage 3 performs inverse projection. In addition, the authors introduce a reranking strategy that prioritizes reactant candidates by their structural similarity to the target product.

**Strengths:**

1. This paper weaves together several established techniques for retrosynthesis—a nontrivial engineering task. Specifically, it integrates molecule–text contrastive learning [1], product–reactant alignment [2], and reranking based on reactant–product similarity [3]. As a result, the work reads less like a single conceptual leap and more like a carefully engineered system.

2. As evidenced in Table 1, augmenting each backbone with the proposed method consistently improves performance. Results in Tables 2 and 3 corroborate this finding, demonstrating that each component is effective and yields measurable contributions.


[1] GeomCLIP: Contrastive Geometry-Text Pre-training for Molecules.
[2] Alignment is Key for Applying Diffusion Models to Retrosynthesis.
[3] Preference Optimization for Molecule Synthesis with Conditional Residual Energy-based Models

**Weaknesses:**

The paper’s novelty is limited: it primarily integrates previously known techniques within one framework. While the system is elaborate, its design lacks conceptual simplicity. Given the largely combinatorial nature of the contribution, the work seems better suited to a lower-tier venue and does not meet the typical standards of ICML/NeurIPS/ICLR. The manuscript is also difficult to follow; the inclusion of many moving parts makes it read more like a technical report than a cohesive research paper.

**Questions:**

N/A

---

### Official Review · Reviewer_oryP · 2025-11-03

**Soundness:** 3
**Presentation:** 2
**Contribution:** 2
**Rating:** 2
**Confidence:** 4

**Summary:**

This paper presents RetroAux: essentially a type of learned nearest neighbor retrieval method to predict additional reactants for a chemical reaction, thereby predicting the reaction set autoregressively. The authors evaluate on USPTO.

**Strengths:**

- The overall idea itself is sensible (learned retrieval methods have been successful in many areas).
- The experimental evaluation is broadly sensible (although I have some concerns, see "weaknesses")
- Writing generally clear, and I liked the figures in the paper!
- Being upfront about limitation that model only works for multi-reactant reactions.

**Weaknesses:**

- **Distasteful exaggerations of paper's novelty/significance** (eg describing the contribution as a "paradigm shift" / "paradigm evolution"). I can't imagine anybody in the field who would agree with this description. If anything, this paper argues for the _existing  paradigm_ of heavier inductive biases in retrosynthesis models. It's always disappointing to read phrases like "paradigm shift" in papers: I don't think it convinces any reviewers to rate the paper any higher (in fact I suspect it has the opposite effect).
- **Limited novelty** all things considered, learned retrieval methods themselves are not very novel. [Retrieval-Retro](https://arxiv.org/abs/2410.21341) did this specifically in the field of retrosynthesis.
- **Concerns with experimental evaluation**:
  - By seeding the dictionary with ground-truth molecules from USPTO, this appears to change the task from purely generative into a discriminative task (aka picking out the ground-truth molecule from a list). I imagine part of the gains in retrosynthesis come from making the task "easier" in this way. If so, it might be good to use discriminative versions of other methods as a baseline (eg evaluating log-likelihoods of different options and picking the top one)
  - Only top-1 accuracy is shown. Why not top 5 or top 10? These are commonly reported in other papers. I would expect the gains to be smaller there.
  - One of the baselines is the original 2017 transformer paper. This was trained primary on English to German translation. I am shocked to see it get ~40% top-1 accuracy on retrosynthesis, a non-natural language task which it was never trained on. Is this a mistake?
- **Some exaggerations of novelty in reaction space**:
  1. On line 447, the authors emphasize the prediction of "three entirely novel reaction pathways" using molecules which were not present in the initial dictionary. I am confused why this is highlighted- basically every model is capable of producing reactions from common classes (eg Suzuki coupling) on novel molecules. While this definitely shows some "generalization capabilities", they are certainly not _exceptional_ (as claimed on line 451-452)
  2. On lines 428-430, the authors claim that retrieving several structurally-related nearest neighbors means that the model "inherently learns fundamental reaction rules, genuinely comprehends complex property correlations between products and reactant". Basically all retrosynthesis models predict clusters of similar reactions (eg Chemformer will predict coupling or protection reactions with a bunch of slightly different reactions in its top ~10 outputs). This is not a capability unique to RetroAux.
- **Motivation / rationale for method unclear**: the motivation in §1 is convoluted: a mix of simplifying generation, putting in inductive biases, and leveraging pre-training data. These all seem like they could be done another way: for example, semi-template methods using foundation model embeddings. It would be nice to understand why the authors specifically recommend RetroAux.

**Questions:**

- Were the ground-truth top 1 molecules always included in the dictionary for evaluation?
- Can you explain the motivation for RetroAux in contrast to other ways of potentially achieving the same goals?

---

### Note · Authors · 2025-12-04

I have read and agree with the venue's withdrawal policy on behalf of myself and my co-authors.